# Oxygen Consumption, Ventilatory Thresholds, and Work Zones in Nordic Walking Competitors

**DOI:** 10.3390/jfmk9030171

**Published:** 2024-09-19

**Authors:** María Serna-Martínez, Sandra Ribes-Hernández, Ignacio Martínez-González-Moro

**Affiliations:** Physical Exercise and Human Performance Research Group, Mare Nostrum Campus, University of Murcia, 30001 Murcia, Spain; mariasernamartinez@gmail.com (M.S.-M.); sandra-ribes@hotmail.com (S.R.-H.)

**Keywords:** exercise physiology, aerobic and anaerobic metabolism, walkers, ambulatory electrocardiography

## Abstract

**Background:** Nordic walking (NW) is a physical sports activity that has been sufficiently studied from the point of view of health, but physiological and performance analyses have not been so much. **Objectives:** With this study, we intend to analyse the physical work areas, according to ventilatory thresholds, that occur during a NW competition. **Methods:** Four participants of different characteristics anthropometrics (weight 57.6–85.6 kg; height 165.8–178 cm; and fat percentage 14.5–21.5%) gender (3 males and 1 female) and age (15–57 years) who participated in the NW regional championship have been chosen, and their electrocardiographic tracing was recorded using a NUUBO^®^ device throughout the race, obtaining average and maximum heart rates (HR) in eight sections of the circuit. Previously, in the laboratory, a maximal stress test was performed to determine the maximum oxygen consumption (VO_2_max), the first (VT1) and second (VT2) ventilatory threshold (VT). With these data, four work areas were obtained. **Results:** Most of the sections of the circuit were conducted with average HRs in zone 2a (above average between VT1 and VT2 but below VT2) and peak HRs in zone 3 (between VT2 and VO_2_max). **Conclusions**: We conclude that, with the data collected on HR, VO_2_max, and VT, the training zones obtained can be related to the heart rates in the different sections of the circuit. This can be used to improve the sports performance of the walkers.

## 1. Introduction

Nordic walking (NW) is an emerging physical-sports activity that is gaining more and more followers, both as a healthy activity and as a competitive activity. NW has its origins in cross-country skiing and basically consists of “walking with ski poles”. It is considered that in 1930 the Finnish National Nordic Ski Team used this technique as training during the summer season, and since 1966, it has been evolving in its technical, didactic, and competitive aspects [1]. In NW, poles specially designed for the development of this physical activity are used [2]. There is a specific technique to propel yourself during movement, maximizing muscle work and range of motion and minimizing overload on the lower joints and enhancing the upper joints. The objective of NW competitions is to complete a circuit in a natural environment in the shortest possible time. The circuits are different for each competition, and they alternate flat, uphill, and descending sections. Thus, the physical and energy requirements are different for each section of the course and for each championship. Since its inception, it has been considered a beneficial activity for health, accessible, and safe [3,4,5]. There are numerous studies carried out on elderly people [6,7] and with various pathologies [8].

In addition to the social [9] and health [5] aspects, the biomechanical aspects of the progression technique [10] and the use of canes have been studied extensively in NW [11] and, to a lesser extent, metabolic [12,13]. It has been seen that, in regular non-competitive NW practitioners, this sport, compared to normal walking, increases oxygen consumption (VO_2_) by an average of 20%, caloric expenditure by 22%, and heart rate response by 16% [14]. When comparing NW with conventional walking (CW), it is seen that NW showed greater work intensity than CW, with an oxygen consumption difference of 1.7 mL·kg^−1^·min^−1^ because NW involved more upper body muscles than CW [15].

There is little research on training and sports performance in competition NW. Very few studies provide data on oxygen consumption [14,15,16], heart rate (HR) [17], or heart rate variability [18]. In NW competitions, athletes do not run; they march by pushing themselves with their poles, reaching speeds higher than those achieved in a conventional walk but lower than those of running [19]. In addition, in NW oxygen consumption, heart rate, expiratory volume, energy expenditure, and lactic acid levels increase at the same subjective rate as when walking [20]. Competitive NW is an aerobic endurance sport in which distances are covered by circuits usually in nature to complete them in the shortest possible time. As in all aerobic endurance sports, the determination of maximal oxygen consumption and ventilatory thresholds 1 and 2 (VT1 and VT2) may be of interest for planning training and improving performance [21].

Ventilatory thresholds (VT) are determined by stress tests carried out in exercise physiology laboratories with the analysis of exhaled gases. After detecting VT, the training areas are described from a metabolic point of view. Its application to training is widely proven in endurance sports such as athletics and long-distance cycling [22,23,24], but not in minority sports practiced in a natural environment such as NW. The energy metabolism during an NW competition may not coincide with that of other endurance sports since for its progression the walker uses poles and moves without running, something like what happens with athletic walking [25].

With this work, we intend to know how an NW competition develops, from the metabolic point of view, based on the knowledge of the heart rate in which the ventilatory thresholds are manifested. NW is an endurance sport, so energy demands are key to achieving success and knowledge of the ventilatory thresholds and training zones necessary for training planning. Therefore, the purpose of this study was to identify the intensity zones of aerobic training in which different sections of a Nordic walking competition are performed (work zones). We hypothesized that the walkers make most of the route in the area near VT2 without exceeding it, regardless of their age and gender.

## 2. Materials and Methods

### 2.1. Design

The study has the authorization of the Research Ethics Commission of the University of Murcia (M10/2023/052). It was carried out following the recommendations of Helsinki, and informed consent was obtained from all participants.

The work was structured in three phases: 1st phase: laboratory stress test and preliminary studies; 2nd phase: Nordic walking competition on natural circuit; and 3rd phase: extraction and analysis of results.

### 2.2. Participants

Four walkers registered in the Regional Nordic Walking Championship of the Region of Murcia (Spain) representing three groups of participants were randomly selected (aleatory numbers; Excel^®^ V16): cadets, female and male veterans; one walker was selected from each group except for the group of male veterans, in which two were selected. A week before the competition, they went to our laboratory for a sport’s medical examination and the performance of a maximum stress test with exhaled gas analysis and the obtaining of the VT1 and VT2 ventilatory thresholds. The walkers went to the competition in their best physical condition. They carried out the conventional training with their trainers, and the researchers have not intervened in their planning. The athletes were asked not to change their nutritional habits between the stress test and the competition. The inclusion criteria were to be registered in the championship and accept the invitation to participate in the study, and the exclusion criteria were to suffer from diseases, injuries, or alterations that contraindicate the performance of the stress test and/or participation in the competition, not to perform a maximum stress test, or not to finish the competition.

### 2.3. Procedure

The characteristics, content, and material used in each phase are as follows:

Phase 1.—After informing each participant of the objectives and procedures of the study, they signed a document with their consent. Subsequently, anthropometric data (weight, height, waist circumference) and body composition (fat percentage and percentage of muscle) were obtained by bioimpedance (Inbody 120, Inbody Co., Seoul, Republic of Korea). Body composition analysis was performed with participants standing, barefoot on the platform, and wearing light clothing during the measurement.

After that, an anamnesis and clinical examination were carried out by a doctor with experience in sports cardiology. Auscultation, blood pressure measurement, decubitus and stress electrocardiogram (Clickecgbt, Cardioline, Cavareno, Italy) and echocardiography (Clarius PA HD3, Clarius, Burnaby, BC, Canada). A standard electrocardiogram of 12 leads in the supine position and an echocardiographic study were conducted with the subjects lying in the left lateral recumbent position, analysing the transthoracic position, in the parasternal position, the long and short axes, and four apical chambers. “M”, “2D”, and “colour doppler” modes were used. This ruled out cardiac abnormalities.

Maximum oxygen consumption (VO_2_ max) and both ventilatory thresholds (VT1 and VT2) were obtained after performing a stress test with an incremental protocol until exhaustion, with exhaled gas analysis. The stress test was performed on a treadmill (runner model run7411, Runner Srl, Cavezzo, Italy). It started with a warm-up of 2 min at 6 km/hour, increasing the speed by one km/hour every minute, keeping the slope stable at 1%. The stress test was carried out in free running, without poles. During the test, oxygen consumption (VO_2_), heart rate (HR), respiratory exchange ratio (RER), pulmonary ventilation (EV), carbon dioxide (VCO_2_) production, partial pressures of O_2_ (%FEO_2_) and CO_2_ (%FECO_2_) exhaled, and ventilatory equivalents for oxygen (EV/VO_2_) and carbon dioxide (EV/VCO_2_) were measured breath-to-breath with the equipment ergospirometry (MetaLyzer 3B, Cortex, Leipzig, Germany).

For each participant, the first (VT1) and second ventilatory threshold (VT2), VO_2_ max, and HR max were determined. For each threshold, the HR and the VO_2_ at which they were produced were noted. The results were analysed by an experienced exercise physiologist who evaluated the changes in the measured parameters with increasing workload. Ventilatory thresholds were determined using the respiratory equivalent method [26]. The first threshold was detected by the rate at which the EV/VO_2_ and FEO_2_ ratio reached a minimum (nadir or first increase in EV/VO_2_ vs. workload without a simultaneous increase in EV/VCO_2_ vs. workload). The second ventilatory threshold was detected by the rate at which the EV/VCO_2_ ratio reached a minimum and the FECO_2_ reached a maximum (nadir or nonlinear increase of EV/VCO_2_ against workload). To determine VO_2_max, the following criteria were applied: plateau in VO_2_, RER > 1.1, and HR within 10 bpm of the maximum predicted by age (220-age). If no plateau was observed but the rest of the criteria were met, the VO_2_peak was taken as VO_2_max [27]. The equipment ergospirometry was calibrated according to the manufacturer’s requirements (gas and volume calibration).

Phase 2.—The competition was held on a natural circuit of 5.55 km in length with a round trip (total length of 11.1 km). All four participants were fitted with a wireless electrocardiogram recorder NUUBO^®^ (Nuubo, Madrid, Spain) attached to a textile harness in contact with their chest (Figure 1). The device consists of five electrodes, two in each fifth intercostal space and another in the sternal manubrium area. To facilitate electrical conduction, the area was cleaned with hydroalcoholic solution, and a little conductive gel (Cardiocream^®^, Nihon Kohden, Rosbach, Germany) was applied. The device continuously collects and records three electrocardiographic channels that are then transmitted to a computer. A few minutes before the starting signal, the recording was activated, and the time was noted. Likewise, the absence of adhesive electrodes and their replacement by a textile harness means that the electrodes do not detach from the skin and that the number of events is low. The recorder and its harness do not impede the movements or gestures typical of this sport.

Five observers were placed in the circuit who noted the time of passage of each walker at their point, establishing a total of eight sections (Figure 2). The cadet runners reached the control of km 3 and turned back; their total distance was 6 km (excluding Section 4 and Section 5). The absolute categories reached the control of km 5.55 and began the return.

3.—Data extraction and analysis. The software used (nECG suite LT v2^®^) extracts the information from the recorder and presents a “minute graph” with the electrocardiographic record of the entire recording. In each row, the beats produced in one minute are recorded, on the left the hour and minute to which they correspond, and on the right the average HR during that minute (Figure 3). Before the data extraction, a detailed viewing is carried out by a specialized doctor to eliminate artifacts and detect “events.” The events are also located automatically by “artificial intelligence”, being revalidated by the researchers. “Events” are arrhythmias (tachycardias, bradycardias, or fibrillations), the appearance of abnormal beats (supraventricular or ventricular extrasystoles), and marks generated by the participant coinciding with clinical symptoms (chest pain, malaise, or dizziness).

For each participant, the exact time of passage through each checkpoint is identified in their “minute graph”, including the start and finish times, differentiating the eight sections of the route (six for the youth). From each section, the average HR, the peak HR, and the valley HR are obtained. The peak and trough HRs are the maximum and minimum obtained, respectively, in a 10-s segment. To do this, the cursor is moved through the “minute graph”, marking segments of 10 s and obtaining the HR during it.

The HRs of each section are transformed into a percentage of the maximum HR obtained in the stress test and are classified according to the exercise zone, considered as the intensity zones of aerobic training.

We have considered four work zones according to the HR. We have considered the three classic zones [28,29] but dividing zone 2 into two subzones [30]. Zone 1 HR is lower than VT1 HR; Zone 2a HR is between HR of VT1, and the median is between VT1 and VT2); Zone 2b is between the median of VT1 and VT2 and the HR of VT2; and Zone 3 HR is greater than VT2 (Figure 4).

Each section of the circuit for each walker has been classified according to the HR zones with which each walker has completed it.

### 2.4. Variables

The heart rates in each of the sections of the circuit and the work areas to which they correspond are described as dependent variables. As independent variables, the HR in which VO_2_ peak, VT1 and VT2 were obtained in the laboratory.

### 2.5. Statistical Analysis

The quantitative variables have been described with the mean values and the standard deviation (SD). The coefficient of variation (VC) has been calculated using the equation VC = SD/mean * 100.

## 3. Results

Table 1 shows the general characteristics (gender, category, age, number of years as walkers and hours per week of NW-specific training), anthropometric values, and body composition of the participants.

Table 2 shows the ergometric results for each of the participants separated by maximum values, VT_1_ and VT_2_. Theoretical maximum heart rate (TMHR) was obtained from the 220-age equation. Based on this value and the HRmax data obtained, the percentage of theoretical HRmax (% TMHR) was calculated.

Based on the HR in which each of the thresholds is reached, the four physical work zones are established for each of the participants (Table 3).

The circuit was divided into eight sections (S1–S8). Table 4 shows the average HR of each section and participant in real values and as a percentage of the maximum HR and the work zones to which they correspond. Figure 5 shows the evolution of the HRmax percentage (of each participant in each section). While in Table 5, we show the peak values of HR in those same sections.

## 4. Discussion

The purpose of this study was to identify the intensity zones of aerobic training in which different sections of a Nordic walking competition are performed (work zones). We have observed that each section of the route can be attributed a work intensity based on the data obtained in a stress test carried out in the laboratory.

We have obtained the heart rates of four walkers during a Nordic walking competition using a NUUBO^®^ wireless recorder. Previously, for each participant, the ventilatory thresholds (VT1 and VT2) were obtained, and the physical work zones were determined according to them, which are also called exercise zones, considered the intensity zones of aerobic training.

The use of a wireless recorder, such as the one we have used in this work, allows total freedom of movement on the part of the participants. The device does not generate any signal and does not interfere with other devices that may be used by participants, other walkers, and organizers (cameras, watches, or devices with GPS). As it is a recorder and the information is not displayed on any screen, it cannot be considered as an unfair aid. For these reasons, we believe that the use of this technique is valid to obtain information about cardiac work during competition. The information extracted from the recorders is more complete than that provided by heart rate monitors. Heart rate monitors only record numerical information about the heartbeat and not the electrocardiographic morphology. The visualization of the electrocardiogram allows events to be ruled out more accurately, so the measurements of HR are more accurate and their clinical usefulness greater.

In sports such as NW, in which you do not run and use poles, creating effort assessment protocols in laboratories is difficult. Therefore, having the help of real information on the field of competition can be of interest to monitor and plan training. This pilot study can be the basis for obtaining data to understand the physical effort of walkers and improve their physical planning and performance.

The VO_2_ max of the veteran walkers in our study is above the values described by Jódar et al. for a group of recreational walkers of similar ages [31] and those cited by Sugiyama 2013 [32]. On the other hand, the VO_2_ values of the women participating in our study (39 mL/kg/min) exceed the mean of those who participated in the study by Wiacek et al. (31.9 mL/Kg/min). [33]; being like the walkers of Jürimaë (43.5 ± 5 mL/kg/min) [16], despite being younger, and to those of Hansen and Smith with 43.4 ± 2.8 mL/kg/min [13].

During the 8 sections of the race, the walkers were always almost in submaximal HRs and in work areas near VT2. These data support the study by Dechman et al. [34], who demonstrated that the physiological responses required to the practice of NM in the natural environment are superior to those studied in the laboratory.

The VT2 anaerobic threshold is a necessary piece of data to set the pace of the race because it indicates the maximum metabolic speed at which lactate can be removed from the muscles that are being exercised, eliminating the CO_2_ that is produced in its metabolization by increasing ventilation. Threshold determination is the best method for defining exercise intensity [35]. Exercise above this threshold produces an accumulation of lactates, which leads to a rapid decrease in performance. Conversely, when an athlete or cyclist can exercise without exceeding VT2, they could maintain that exercise intensity for an indefinite time if energy intake is maintained [36]. The intensity with which Nordic walking competitions are practiced is similar to that of other outdoor sports such as cross-country skiing [37] and triathlon [38]. The walkers in our study have mostly remained in zone 2b during the competition, therefore, close to the anaerobic threshold, although they have presented HR peaks close to the maximum heart rate and even exceeding it at certain times, entering zone 3. The percentage of HR with respect to the maximum in our walkers has ranged from 75.6 of young runners to 92.2 and 94.4% of veterans, percentages like those shown by Ronconi and Alvero-Cruz [39] in all segments in a duathlon competition. In the young walker, the chances of improving their performance are greater. A factor that can condition a lower performance in the cadet category athlete may be the specific technical aspects of the competition. Unlike other sports, in NW there are judges during the course who penalize non-regulation techniques. This influences young athletes and conditions physical performance. It also happens with the technique of handling the pole [10]. On uneven terrain it can be a limiting factor of maximum effort.

As has been seen in other studies [15], the use of poles helps progression and increases speed but does not contribute to reducing energy demands. In this work, we have seen that in NW VO_2_ values close to the maximum are reached. Thus, we consider that the sporting gesture and marching instead of running affect the final speed but not the consumption of oxygen. In NW, upper and lower limbs move and exercise, which makes the energy demand greater than normal walking. The speed obtained in the NW competition is lower than the maximum speed obtained in the laboratory, and the characteristics of the terrain (ramps and slopes) increase the energy requirements to maintain speed.

Knowing the relationship between the effort made, HR, and ventilatory thresholds allows us to improve training systems and improve performance in endurance sports on natural terrain [15]. In these sports, the demands are different from track or stadium competitions. The peculiarities of the routes change from competition to competition, and their rhythms and characteristics are difficult to reproduce in the laboratory. Coaches can analyse the HR zones developed according to the characteristics of the route and guide their athletes on how to dose the effort to improve results.

This pilot study allows us to establish a line of research that covers not only NW practitioners but also trail athletes, orienteering races, and in general off-piste endurance sports. It can be a way to personalize the training of future competitions with real information, relating the results in the laboratory with those obtained outdoors.

The main limitation is the number of recording devices that can be used simultaneously to compare groups of participants. Another difficulty is the comparison between competitions since each one is held with a different route. On the other hand, the laboratory stress test has been carried out without poles and with a running protocol, which makes it mechanically different from outdoor running. It might be interesting to use protocols with poles, as Giovanelli et al. [40] recently did. We do not consider it essential since they did not obtain differences between VO_2_ max values nor in maximum heart rates or ventilatory thresholds, although they did find differences in the perception of effort.

## 5. Conclusions

This study has made it possible to determine the usefulness of the data obtained in the laboratory together with the continuous electrocardiographic recording in the physical assessment of walkers for training planning.

We conclude that, with the data collected on heart rate, oxygen consumption, and ventilatory thresholds, the training zones obtained can be related to the heart rates in the different sections of the routes. This can be used to improve the sports performance of the walkers. NW can be considered a high-intensity sport performed in free circuits.

## Figures and Tables

**Figure 1 jfmk-09-00171-f001:**
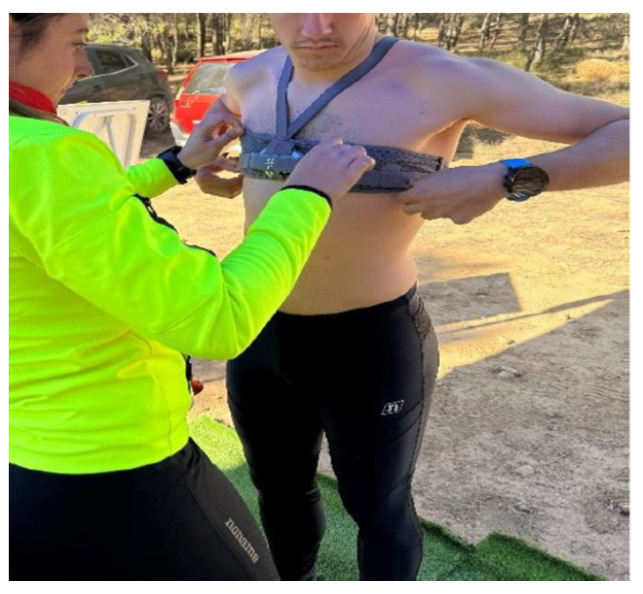
Placement of the Nuubo^®^ device.

**Figure 2 jfmk-09-00171-f002:**
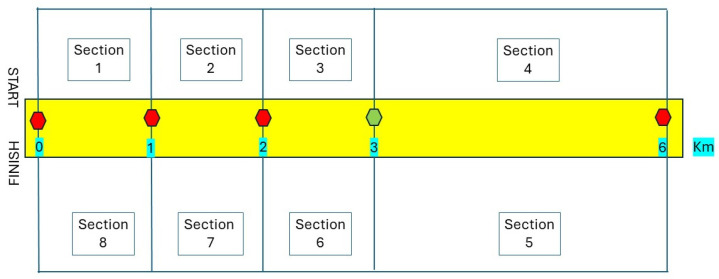
Diagram of the circuit, controls, and sections. The red and green dots indicate the location of the time controls. The green dot marks the return of the cadet category.

**Figure 3 jfmk-09-00171-f003:**
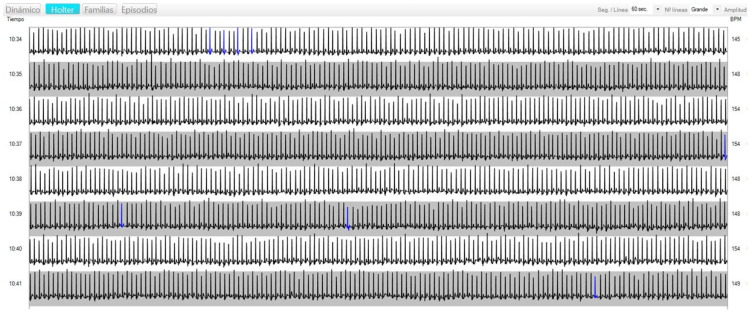
Example of minute electrocardiogram graph. The registration between 10.34 a.m. and 10.41 a.m. displayed. Blue shows events, in this case several supraventricular extrasystoles at minutes 34, 37, 39, and 41.

**Figure 4 jfmk-09-00171-f004:**
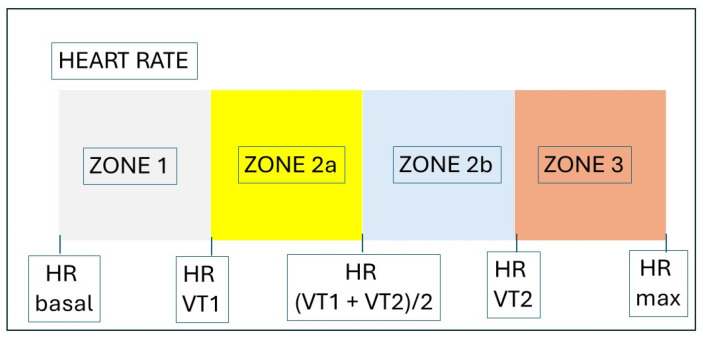
Heart rate (HR) zones from ventilatory thresholds (VT1 = first ventilatory threshold; VT2 = second ventilatory threshold).

**Figure 5 jfmk-09-00171-f005:**
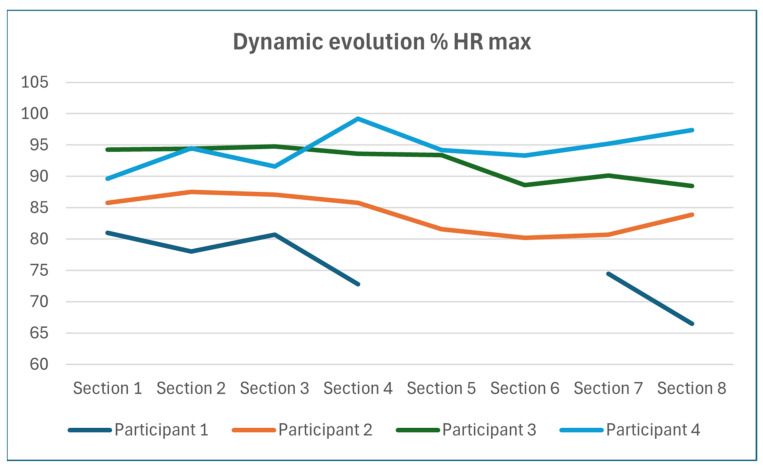
Dynamic evolution % HR max.

**Table 1 jfmk-09-00171-t001:** Characteristics of the participants.

	PARTICIPANTS						
	Nº 1	Nº 2	Nº 3	Nº 4	Mean	SD	VC (%)
Gender	Male	Female	Male	Male			
Category	Cadet	Veteran	Veteran	Veteran			
Aged (Years)	15	55	57	56	45.75	17.77	44.84
Years NW	2	8	6	5	5.25	2.17	47.62
Weekly hours	6	8	8	7	7.3	0.96	13.20
Weight (Kg)	60.2	57.6	84.2	85.6	71.9	13.04	20.94
Height (cm)	175.5	165.8	178	176	173.8	4.73	3.14
Waist circumference (cm)	72	74	92	94	83	10.05	13.98
Hip circumference (cm)	90	92	97	99	94.5	3.64	4.45
BMI (Kg/m^2^)	19.55	20.95	26.57	27.63	23.675	3.48	16.98
Fat percentage (%)	14.5	21.5	20	17.4	18.35	2.66	16.76

**Table 2 jfmk-09-00171-t002:** Stress test data.

PARTICIPANTS	Nº 1	Nº 2	Nº 3	Nº 4	Mean	SD	CV
TMHR (bbm)	205	165	163	164	174.3	20.5	11.8
HR max (bbm)	208	178	151	163	175.0	24.6	14.1
% TMHR	101	107	92	99	99.8	6.2	6.2
RER	1.14	1.12	1.14	1.11	1.1	0.0	1.3
VO_2_ peak (mL/Kg/min)	58	39	37	46	45.0	9.5	21.1
Maximum ventilation (L/min)	125	65.7	101	150	110.4	35.9	32.5
Maximum velocity (Km/h)	16	13.3	13.8	13.1	14.1	1.3	9.5
HR In VT1 (bbm)	160	132	119	127	134.5	17.8	13.3
% HR max in VT1	76	74	78	78	76.5	1.9	2.5
VO_2_ in VT1 (mL/Kg/min)	38	27	25	28	29.5	5.8	19.7
% VO_2_ max in VT1	65	69	67.6	61	65.7	3.5	5.4
VelocIty in VT1 (Km/h)	9.9	8.2	9	8.1	8.8	0.8	9.5
HR in VT2 (bbm)	200	165	143	157	166.3	24.3	14.6
% HR max in VT2	96	92	94	96	94.5	1.9	2.0
VO_2_ in VT2 (mL/Kg/min)	56	38	35	44	43.3	9.3	21.5
% VO_2_ max in VT2	96	97	94	95.6	95.7	1.2	1.3
Velocity in VT2 (Km/h)	14.7	12	12.3	12.6	12.9	1.2	9.5

TMHR: Theoretical maximum heart rate. Maximum teoric. SD: Standard deviation. CV: Coefficient of variation. HR: heart rate. RER: respiratory exchange ratio. VO_2_: Oxygen uptake. VT1: first ventilatory threshold. VT2: second ventilatory threshold. bbm: Beat by minute. Max: maximum.

**Table 3 jfmk-09-00171-t003:** Working zones according to heart rate (bbm) at ventilatory thresholds.

Participant	ZONE1 <HR VT1	ZONE 2a HR VT1 − (VT1 + VT2)/2	ZONE 2b (VT1 + VT2)/2 − HR VT1	ZONE 3 >HR VT2
Nº 1	<160	160–180	181–200	>200
Nº 2	<132	132–148	149–165	>165
Nº 3	<119	119–131	132–143	>143
Nº 4	<127	127–139	140–157	>157

**Table 4 jfmk-09-00171-t004:** HR means by sections and work zones.

HR Mean	Nº	S 1	S 2	S 3	S 4	S 5	S 6	S 7	S 8	Means ± SD
HR mean (bbm)	1	168.5	162.2	167.9	151.5	*	*	154.9	138.3	157.2 ± 11.5
	2	152.7	155.7	155.1	152.7	145.2	142.8	143.7	149.4	149.7 ± 5.2
	3	142.4	142.6	143.2	141.4	141	133.8	136	133.6	139.3 ± 4.1
	4	146.1	154	149.3	161.7	153.5	152	155.1	158.7	153.8 ± 4.9
% HR max	1	81.0	78.0	80.7	72.8	*	*	74.5	66.5	75.6 ± 5.5
	2	85.8	87.5	87.1	85.8	81.6	80.2	80.7	83.9	84.1 ± 2.9
	3	94.3	94.4	94.8	93.6	93.4	88.6	90.1	88.5	92.2 ± 2.7
	4	89.6	94.5	91.6	99.2	94.2	93.3	95.2	97.4	94.4 ± 3.0
Work Zones	1	Z2a	Z2a	Z2a	Z1	*	*	Z1	Z1	
	2	Z2b	Z2b	Z2b	Z2b	Z2a	Z2a	Z2a	Z2b	
	3	Z2b	Z2b	Z2b	Z2b	Z2b	Z2b	Z2b	Z2b	
	4	Z2b	Z2b	Z2b	Z3	Z2b	Z2b	Z2b	Z3	

Nº: Number participant. HR; Heart rate. SD: Standard deviation. S: sections. bbm: beat by minute. * It corresponds to the sections that were not part of its route.

**Table 5 jfmk-09-00171-t005:** HR peaks by sections and work zones.

HR Peak	Nº	S 1	S 2	S 3	S 4	S 5	S 6	S 7	S 8	Means ± SD
HR (bbm)	1	185	178	192	160	*	*	189	151	175.8 ± 16.7
	2	161	162	170	170	166	156	150	156	161.4 ± 7.2
	3	147	148	146	147	147	140	142	138	144.4 ± 3.8
	4	158	157	161	165	161	157	161	163	160.4 ± 2.9
% HR max	1	88.9	85.6	92.3	76.9	*	*	90.9	72.6	84.5 ± 8.0
	2	90.4	91.0	95.5	95.5	93.3	87.6	84.3	87.6	90.7 ± 4.0
	3	97.4	98.0	96.7	97.4	97.4	92.7	94.0	91.4	95.6 ± 2.5
	4	96.9	96.3	98.8	101.2	98.8	96.3	98.8	100.0	98.4 ± 1.8
Work zones	1	Z2b	Z2a	Z2b	Z2a	*	*	Z2b	Z2a	
	2	Z2b	Z2b	Z3	Z3	Z3	Z2b	Z2b	Z2b	
	3	Z3	Z3	Z3	Z3	Z3	Z2b	Z2b	Z2b	
	4	Z3	Z3	Z3	Z3	Z3	Z3	Z3	Z3	

Nº: Number participant. HR: Heart rate. SD: Standard deviation. S: sections. bbm: beat by minute. *: Section without measurements.

## Data Availability

The data can be requested by e-mail from the corresponding author.

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
