# Peer review of "Oxygen Consumption, Ventilatory Thresholds, and Work Zones in Nordic Walking Competitors"

_jfmk, 2024, doi:10.3390/jfmk9030171_

Round 1

Reviewer 1 Report

Comments and Suggestions for Authors

13. You can use the participants’ anthropometrics

24. Differentiate the keywords from the title.

43. Please, check if the percentages, in the used literature, are correct.

49 You can include the percentage of difference.

90. Had they any specific preparation before their race? Were they in the best condition for the upcoming race? Did you check their daily nutritional habits?

132. This should be a probable limitation of your study. If someone has measured according to his/her VO2 peak or max.

151. Figure 2 is a nice idea; however, I suggest explaining what these sections mean and specifying the colored bullets.

182. Zone 1:, Zone 2:, etc. Also, a blood lactate analysis would enhance your Zones specification.

247. Specify the * in the table.

256. It is better to begin with the study’s general results and continue with a more specific discussion.

261-274. I cannot understand the purpose of this paragraph. I expect to read about the physiological burden of Nordic walking, not the tool's technical points.

295. lactate

310. Any possible explanation?

318. Delete the full stop.

The discussion must be modified at some points. Also, I suggest comparing it with other sports with similar physiological burdens, expanding your debate and discussion.

Author Response

We thank the reviewer for the time spent on our work and their feedback.

1 Not clear how characterizing the metabolic and physiological demands of NW could be an added value from the introduction

We appreciate the reviewer's comment. NW is an endurance sport, so energy demands are key to achieving success and knowledge of the ventilatory thresholds and training zones necessary for training planning.

We have included a new sentence, in the introduction, clarifying this point.

2 Not clear how the authors expect “that the walkers make most of the route in the area near VT2 without exceeding it, regardless of their age and gender” based on their theorerical background.

Zone nomenclature has been changed to make them more understandable.

3 Is it stress test a proper definition of a classic incremental test to exhaustion protocol? Could running in this protocol be considered appropriate to characterize NW activity? 

We consider the name to be correct. For Nordic walking, it would theoretically be more suitable to perform the tests with poles. Previous experience indicates that using poles on a treadmill is difficult and dangerous, so we reject that possibility. As our work is not based on biomechanical aspects, we believe that it is not a problem to run the test freely. We look for a value of VO2 max and HR max for each subject to relativize the other data.

4 These issues (both, the points raised in the introduction and in the methods) must be considered and more extensive explanations included in order to increase the value of the presented data.

We appreciate the reviewer's comments and have included changes in the different sections to improve the work.

13 you can use the participants’ anthropometrics

              Thanks for the suggestion. We have included these data in the summary.

  1.  Differentiate the keywords from the title.

              Thanks for the suggestion. We've changed the keywords.

Exercise physiology, aerobic and anaerobic metabolism, walkers, ambulatory electrocardiography

  1.  Please, check if the percentages, in the used literature, are correct.
  2.  Had they any specific preparation before their race? Were they in the best condition for the upcoming race? Did you check their daily nutritional habits?

              The walkers went to the competition in their best physical condition. They carried out the conventional training with their trainers and the researchers have not intervened in their planning. The walkers were asked not to change their nutritional habits between the stress test and the competition. We have included the above aspects in the text.

  1.  This should be a probable limitation of your study. If someone has measured according to his/her VO2 peak or max.

              To unify criteria, we modified the concept of VO2max to VO2Peak in the results.

  1. Figure 2 is a nice idea; however, I suggest explaining what these sections mean and specifying the colored bullets.

              We appreciate the suggestion of the reviewer and add the information in the caption of the figure. “The red and green dots indicate the location of the time controls. The green dot marks the return of the cadet category”.

182. Zone 1:, Zone 2:, etc. Also, a blood lactate analysis would enhance your Zones specification.

              We appreciate the suggestion, but in this work we have not performed lactate analyses. It was not possible to stop the walkers to obtain the samples.

  1. Specify the * in the table.

Agradecemos la sugerencia e incluimos la explicación. * = It corresponds to the sections that were not part of its route.

  1. It is better to begin with the study’s general results and continue with a more specific discussion.

Hemos incluido un párrafo más general. “The purpose of this study was to identify the intensity zones of aerobic training in which different sections of a Nordic walking competition are performed (work zones). We have observed that each section of the route can be attributed a work intensity based on the data obtained in a stress test carried out in the laboratory”.

261-274. I cannot understand the purpose of this paragraph. I expect to read about the physiological burden of Nordic walking, not the tool's technical points.

En nuestro trabajo, son igualmente importantes los datos obtenidos y la forma en la que se han obtenido. Hemos querido destacar que la tecnología empleada hace posible que los datos se puedan obtener. No hay experiencia en colocar dispositivos de electrocardiografía ambulatoria en competiciones de marcha nórdica.

  1.  lactate

Thank you. We fixed the bug.

  1.  Any possible explanation?

We thank the reviewer for the comment and have expanded the paragraph to include an explanation.

In NW, upper and lower limbs move and exercise, which makes the energy demand greater than normal walking. The speed obtained in the NW competition is lower than the maximum speed obtained in the laboratory and the characteristics of the terrain (ramps and slopes) increase the energy requirements to maintain speed.

  1. Delete the full stop.

 Thank you. We fixed the bug.

The discussion must be modified at some points. Also, I suggest comparing it with other sports with similar physiological burdens, expanding your debate and discussion.

In accordance with the suggestions of all the reviewers, we have modified and expanded some aspects of the discussion. We consider that the comparison of numerical heart rate values between our athletes and those of other studies is of little interest. Our data are from individual walkers, we do not have representative data for a population. Therefore, comparing a subject with a group of different characteristics is not important.

Reviewer 2 Report

Comments and Suggestions for Authors

Title

I do not think that this is a pilot study. It is more a exploratory research.

Abstract

Provide sex and age for each participant.

Introduction

There is a need to provide a background for NW competition. When did NW competitions begin? What are the basic characteristics of a NW competition?

I suggest erasing the hypothesis. Otherwise, the authors should provide a solid background for stating this hypothesis and the reasons behind it.

Methods

How the participants were recruited?

Why the authors recruited two veterans?

Who did perform all assessments?

Please, clarify whether the initial stress tests were performed by walking with poles.

If the protocol described in phases 2 and 3 have been used by the authors themselves or by other authors in previous investigations, it should be accurately referenced.

Line 182 On which bases were the four zones stablished?

Results

This section would benefit from the inclusion of figures indicating HR dynamics during the NW competition.

Theoretical HR max is assessed by the Fox et al equation (220-age). Nevertheless, this equation has been questioned by some authors doi: 10.1590/s0066-782x2008001700005, while backed by some others: PMC7523886.

Since participants show different characteristics, mainly age and sex, the authors need to justify the use of an accurate equation for estimating HR max in each case.

Discussion

Lines 261-275. This information should be provided in methods, as stated earlier (accuracy of the protocol used in phases 2 and 3).

Lines 281-286, discuss VO2 values for each participant, not only for veterans and for women.

Based on the gathered data, can be NW competition considered as high demanding sport? Which other sports modalities have yielded similar effort levels?

NW competitors must keep a specific technique and movement pattern during the competition so as not to be penalized. Could this affect the possibility of reaching a maximal effort? Please, comment.

Lines 323…The main limitation is the sample size, obviously.

Conclusion

This section does not provide useful information. Indicate valid data (HR, VO2, HR work zones) and state whether NW competition can be considered a high effort demanding sport.

Comments on the Quality of English Language

Some sentences could be shorter.

Author Response

We thank the reviewer for the time spent on our work and their feedback.

1.- I do not think that this is a pilot study. It is more a exploratory research.

We thank the reviewer for his comment. We've changed the title

Abstract

2.- Provide sex and age for each participant.

Thanks for the suggestion. We've modified the text and added the maximum and minimum data.

Introduction

3.- There is a need to provide a background for NW competition. When did NW competitions begin? What are the basic characteristics of a NW competition?

Thanks for the comment. We have included a paragraph related to the characteristics of the NW.

4.- 0I suggest erasing the hypothesis. Otherwise, the authors should provide a solid background for stating this hypothesis and the reasons behind it.

              We believe that with the current wording the hypothesis can be maintained.

Methods

5.-How the participants were recruited?

Participants were recruited from the registrant lists and reordered using the Excel random numbers option. The first of each group were invited to participate

6.- Why the authors recruited two veterans? 

Because it was the largest group. It has been indicated in the text.

From the largest group (veteran males), two participants were selected.

7.-Who did perform all assessments?

The evaluations were carried out by a doctor specializing in sports medicine and expert in exercise physiology (IMG-M) lines 86 y 123

8.- Please, clarify whether the initial stress tests were performed by walking with poles.

Thanks for the observation. The initial tests were carried out running, without poles. We clarify this in the procedure section.

9.- If the protocol described in phases 2 and 3 have been used by the authors themselves or by other authors in previous investigations, it should be accurately referenced.

As the reviewer knows, training zones are usually divided into three, using the heart rate values of the ventilatory thresholds as interval limits. There is a tendency to divide zone 2 into two subzones, each close to a threshold which is what we have used. We have changed the nomenclature of our zones so as not to cause confusion. We include references to justify this.

10.- Line 182 On which bases were the four zones stablished? We have explained it with the previous comment

Results

11.- This section would benefit from the inclusion of figures indicating HR dynamics during the NW competition.

              We have included Figure 5 showing the evolution of the HRmax percentage

12.- Theoretical HR max is assessed by the Fox et al equation (220-age). Nevertheless, this equation has been questioned by some authors doi: 10.1590/s0066-782x2008001700005, while backed by some others: PMC7523886. Since participants show different characteristics, mainly age and sex, the authors need to justify the use of an accurate equation for estimating HR max in each case.

              We agree with the reviewer on the questionability of this equation. But that's the most widely used equation, and the one that our gas analyzer's software calculates automatically. In any case, this value is irrelevant, we only use it as another reference to determine that the stress test has been maximum. For the calculations of the work, the real values have been used.

Discussion

  1. Lines 261-275. This information should be provided in methods, as stated earlier (accuracy of the protocol used in phases 2 and 3).

We reformed part of the paragraph and moved information to the other section.

14.- Lines 281-286, discuss VO2 values for each participant, not only for veterans and for women.

We have not found values of the cadet walkers. That is why they are not cited.

15.- Based on the gathered data, can be NW competition considered as high demanding sport? Which other sports modalities have yielded similar effort levels?

Yes, it can be considered. We have completed the comparisons including a related phrase.

17.- NW competitors must keep a specific technique and movement pattern during the competition so as not to be penalized. Could this affect the possibility of reaching a maximal effort? Please, comment.

We think so. Unlike other sports, in Nordic walking there are judges during the course who penalize non-regulation techniques. This influences young athletes and conditions physical performance. It also happens with the technique of handling the cane, on uneven terrain it can be a limiting factor.

We welcome the reviewer's comment and include a new paragraph in the discussion.

18.- Lines 323…The main limitation is the sample size, obviously.

Conclusion

This section does not provide useful information. Indicate valid data (HR, VO2, HR work zones) and state whether NW competition can be considered a high effort demanding sport.

We have modified the wording. But we do not provide numerical data in the conclusion because we have obtained them from few subjects and there is no statistical support to be able to generalize them.

Reviewer 3 Report

Comments and Suggestions for Authors

Thanks for the opportunity or revising the present work.

The manuscript present an interesting idea even if in the form of a preliminary case study. However, several points must be better clarified.

Here below my main comments.

Not clear how characterizing the metabolic and physiological demands of NW could be an added value from the introduction

Not clear how the authors expect “that the walkers make most of the route in the area near VT2 without exceeding it, regardless of their age and gender” based on their theorerical background.

Is it stress test a proper definition of a classic incremental test to exhaustion protocol? Could running in this protocol be considered appropriate to characterize NW activity? 

These issues (both, the points raised in the introduction and in the methods) must be considered and more extensive explanations included in order to increase the value of the presented data.

Author Response

We thank the reviewer for the time spent on our work and their feedback.

1 Not clear how characterizing the metabolic and physiological demands of NW could be an added value from the introduction

We appreciate the reviewer's comment. NW is an endurance sport, so energy demands are key to achieving success and knowledge of the ventilatory thresholds and training zones necessary for training planning.

We have included a new sentence, in the introduction, clarifying this point.

2 Not clear how the authors expect “that the walkers make most of the route in the area near VT2 without exceeding it, regardless of their age and gender” based on their theorerical background.

Zone nomenclature has been changed to make them more understandable.

3 Is it stress test a proper definition of a classic incremental test to exhaustion protocol? Could running in this protocol be considered appropriate to characterize NW activity? 

We consider the name to be correct. For Nordic walking, it would theoretically be more suitable to perform the tests with poles. Previous experience indicates that using poles on a treadmill is difficult and dangerous, so we reject that possibility. As our work is not based on biomechanical aspects, we believe that it is not a problem to run the test freely. We look for a value of VO2 max and HR max for each subject to relativize the other data.

4 These issues (both, the points raised in the introduction and in the methods) must be considered and more extensive explanations included in order to increase the value of the presented data.

We appreciate the reviewer's comments and have included changes in the different sections to improve the work.

Round 2

Reviewer 1 Report

Comments and Suggestions for Authors

Please, answer my comment in English

261-274. I cannot understand the purpose of this paragraph. I expect to read about the physiological burden of Nordic walking, not the tool's technical points.

En nuestro trabajo, son igualmente importantes los datos obtenidos y la forma en la que se han obtenido. Hemos querido destacar que la tecnología empleada hace posible que los datos se puedan obtener. No hay experiencia en colocar dispositivos de electrocardiografía ambulatoria en competiciones de marcha nórdica.

Comments on the Quality of English Language

A whole manuscript check is mandatory.

Author Response

We ask the reviewer to excuse our error.

I enclose the text in English:

261-274. I cannot understand the purpose of this paragraph. I expect to read about the physiological burden of Nordic walking, not the tool's technical points.

In our work, the data obtained and the way in which they have been obtained are equally important. We wanted to emphasize that the technology used makes it possible for the data to be obtained. There is no experience in placing ambulatory electrocardiography devices in Nordic walking competitions.

Reviewer 2 Report

Comments and Suggestions for Authors

No further comments. 

Author Response

Many thanks to the reviewer.

Reviewer 3 Report

Comments and Suggestions for Authors

Thank you to the authors for incorporating my suggestions, which have made the work clearer overall, though I believe there are still a few areas that could be improved. However, these are minor issues.

For instance, in response to my query about the definition of the incremental test, the authors simply stated, "We consider the name to be correct," without providing further explanation or context. While this is indeed a minor point, I was expecting a more detailed justification, as I was for a few other points raised.

That said, I have no further major comments and wish the authors the best of luck with the next steps of the revision process.

Author Response

We thank the reviewer for his comment and for their wishes.

In line 111 of the second version and 121 of the third we expounded: a stress test, until exhaustion. Then in the 113 and 114 (123-124 of the third version) it is indicated: it started with a warm-up of 2 minutes at 6 km/hour, increasing the speed by one km/hour every minute, keeping the slope stable at 1%. 

We believe that the definition proposed by the reviewer is included. In any case, according to your interest, we include a slight change so that your suggestion is collected:

… a stress test, with an incremental protocol, until exhaustion…